# Depletion of Gibberellin Signaling Up-Regulates *LBD16* Transcription and Promotes Adventitious Root Formation in *Arabidopsis* Leaf Explants

**DOI:** 10.3390/ijms252413340

**Published:** 2024-12-12

**Authors:** Tingting Jing, Qian Xing, Yunfeng Shi, Xuemei Liu, Ralf Müller-Xing

**Affiliations:** 1College of Life Science, Northeast Forestry University, Harbin 150040, Chinaliuxuemei@nefu.edu.cn (X.L.); 2Jiangxi Provincial Key Laboratory of Plant Germplasm Innovation and Genetic Improvement, Lushan Botanical Garden, Chinese Academy of Sciences, Jiujiang 332900, China; qxing@lsbg.cn; 3Plant Epigenetics and Development, Lushan Botanical Garden, Chinese Academy of Sciences, Nanchang 330114, China; 4College of Life Science, Nanchang University, Nanchang 330047, China; 5College of Life Science, South China Normal University, Guangzhou 510631, China

**Keywords:** adventitious roots, de novo root regeneration (DNRR), paclobutrazol (PBZ), blocking of gibberellin biosynthesis, rescuing difficult-to-root mutant leaf explants, *LATERAL ORGAN BOUNDARIES DOMAIN 16* (*LBD16*), *LATERAL ROOT PRIMORDIUM 1* (*LRP1*)

## Abstract

Adventitious root (AR) formation in plants originates from non-root organs such as leaves and hypocotyls. Auxin signaling is essential for AR formation, but the roles of other phytohormones are less clear. In *Arabidopsis*, at least two distinct mechanisms can produce ARs, either from hypocotyls as part of the general root architecture or from wounded organs during de novo root regeneration (DNRR). In previous reports, gibberellin acid (GA) appeared to play reverse roles in both types of ARs, since GA treatment blocks etiolation-induced AR formation from hypocotyls, whereas GA synthesis and signaling mutants apparently displayed reduced DNRR from detached leaves. In order to clarify this contradiction, we employed the GA biosynthesis inhibitor paclobutrazol (PBZ) and found that PBZ had positive effects on both types of AR formation in *Arabidopsis.* Consistently, GA treatment had negative effects on both AR formation mechanisms, while loss of GA synthesis and signaling promoted DNRR under our conditions. Our results show that PBZ treatment can rescue declined AR formation in difficult-to-root leaf explants such as *erecta* receptor mutants. Furthermore, transcriptional profiling revealed that PBZ treatment altered GA, brassinosteroids, and auxin responses, which included the up-regulation of *LBD16* that is well known for its pivotal role in AR initiation.

## 1. Introduction

Adventitious roots (ARs) originate, by definition, from non-root tissues and organs. There are different types of ARs, and their overall function, form, and placement vary depending on the species and their individual needs [1]. In the grass family (Poaceae), AR formation often builds up the main part of the root system architecture, whereby the crown roots of maize and rice, which originate secondarily from the stem, are the best investigated examples of ARs in monocots [2]. Members of the plant-specific LATERAL ORGAN BOUNDARIES DOMAIN (LBD) transcription factor family are key regulators of the post-embryonic shoot-borne crown root initiation process in rice and maize [3,4,5,6]. Beside grafting [7,8], AR formation from stem cuttings is the most common method for asexual plant propagation in forestry and agriculture. In the model plant *Arabidopsis*, there are at least two types of AR formation, namely wounding-induced de novo root regeneration (DNRR) from detached organs such as leaves and etiolation-induced AR formation in intact hypocotyls [9]. *LBD16* is a key regulator of auxin-induced callus induction, lateral root (LR) initiation, and both types of AR formation, and it works downstream of AUXIN RESPONSE FACTORs (ARFs) such as ARF7/19 and ARF6/8 [10,11,12,13,14,15]. The loss of *LBD16* results in reduction of DNRR, LR initiation, and AR formation from hypocotyls, demonstrating the general role of *LBD16* in initiating secondary roots [13].

Plant development relies on stem cells harbored in meristems and the correct patterning of cell fates during organogenesis [16]. Adventitious rooting requires the accumulation of auxin in competent vasculature-associated pluripotent cells (VPCs) to activate ARFs, leading to the fate transition of these cells to AR founder cells that establish de novo root apical meristems (RAMs) including root stem cells [17,18,19]. Therefore, auxin biosynthesis and transport play a key role in DNRR in *Arabidopsis* leaf explants [20,21]. In plants, indole-3-acetic acid (IAA) is the main auxin that is primarily synthesized via two chemical reactions [22]. First, the TRYPTOPHAN AMINOTRANSFERASE OF ARABIDOPSIS (TAA) family of amino transferases converts the amino acid tryptophan to indole-3-pyruvate (IPA), which is followed by the conversion of IPA to IAA by the catalysis of flavin-containing monooxygenases encoded by *YUCCA* (*YUC*) genes [23,24]. Several *YUC* genes contribute to AR formation from *Arabidopsis* leaf explants, while *YUC9* expression is primarily induced in response to darkness, and *yuc9* single mutants have a rooting defect in the dark but not in the light [20]. To trigger earlier steps of auxin biosynthesis in detached *Arabidopsis* leaves, wounding-induced jasmonate signaling activates the *ETHYLENE RESPONSE FACTOR* genes *ERF109* and *ERF111*, which, in turn, up-regulates the auxin biosynthesis gene *ANTHRANILATE SYNTHASE α1* (*ASA1*) [25,26]. To prevent hypersensitivity to the wounding signaling, SQUAMOSA PROMOTER BINDING PROTEIN-LIKE (SPL) transcription factors SPL10 and SPL11 and the epigenetic regulator ULTRAPETALA1 (ULT1) attenuate the transcriptional induction of *ERF109*, and partially other ERFs, while JASMONATE-ZIM-DOMAIN (JAZ) proteins inhibit ERF109 activity by direct protein–protein interaction two hours after detachment of the leaf [25,26,27]. Therefore, jasmonate serves as a transient wound signal that triggers auxin biosynthesis, which is required for forming the auxin maximum that stimulates the fate transition of VPCs into AR founder cells. In contrast, jasmonate serves as a negative regulator of etiolation-induced AR formation from *Arabidopsis* hypocotyls by crosstalk with cytokinin [28]. The jasmonate signaling pathway inhibits AR initiation from hypocotyls by repressing of *CYTOKININ OXIDASE/DEHYDROGENASE 1* (*CKX1*) that encodes the CK-degrading enzyme that decreases the cytokinin content during adventitious rooting [1,28]. Accordingly, jasmonate plays opposite regulatory roles in both types of AR formation at different stages and from different sources through crosstalk with other hormones [1].

The phytohormone gibberellic acid (GA) regulates several key processes in plants that are of significant agricultural importance, including root and shoot development, flowering, seed germination, and general plant growth [29]. During GA biosynthesis, various precursors and non-bioactive GAs act as intermediates in the conversion to bioactive GAs catalyzed by a group of enzymes, including ENT-COPALYL DIPHOSPHATE SYNTHETASE 1/GA REQUIRING 1 (CPS/GA1) and GA 20-oxidases such as GA20ox1/GA5 [30]. GA binds to the soluble receptor protein GIBBERELLIN INSENSITIVE DWARF 1 (GID1) and functions as a molecular glue to promote the formation of the GID1–DELLA complexes, which subsequently recruits the ubiquitin E3 ligase SLEEPY1 to trigger polyubiquitination and degradation of DELLA proteins [31]. As the primary target of GA signaling, DELLA proteins modulate gene expression by interacting with other transcription factors, including PHYTOCHROME INTERACTING FACTORs (PIFs) [32,33], BRASSINAZOLE RESISTANT 1 (BZR1) [34,35], JAZs causing the release of MYC2 [36], ARFs [37], and SPLs [38,39]; and other chromatin-associated proteins such as histone H2A [40].

GA is widely recognized to inhibit AR formation from tissue cultures, stem cuttings, and leaf explants in a large variety of plant species [41,42,43,44,45,46,47,48]. The few studies that reported increased AR formation by GA treatment described likely indirect effects such as promotion of preexisting root anlagen or GA stimulating AR formation via a mechanism that requires the presence of ethylene [41,49]. Nevertheless, a recent study indicated the requirement of GA and tight regulation of GA signaling through DELLA proteins such as GIBBERELLIC ACID INSENSITIVE (GAI) to promote wounding-induced AR formation in *Arabidopsis* leaf explants [50]. This conclusion is based on the phenotype analysis of several classical and newly identified GA synthesis and signaling mutants in DNRR assays, which did not include GA or other phytohormone treatments [50]. In contrast, GA treatment strongly inhibits etiolation-induced AR formation from hypocotyl [43]. These contrary outcomes might have been caused by different mechanisms involved in both types of AR formation or resulted from the different methods that were used in both studies. 

The plant growth retardant paclobutrazol (PBZ, PAC, PP333) [(2RS,3RS)-1-(4-chlorophenyl)-4,4-dimethyl-2-(1,2,4-triazol-1-yl)pentan-3-ol] specifically inhibits the three steps in the oxidation of the GA-precursor ent-kaurene to ent-kaurenoic acid catalyzed by ENT-KAURENE OXIDASE/GA REQUIRING 3 (KO/GA3) [51,52,53]. In *Arabidopsis*, PBZ treatment has frequently been employed as GA biosynthesis inhibitor to investigate the role of GA signaling during plant development [54,55,56,57,58,59,60], but PBZ has not been previously used for *Arabidopsis* AR formation studies. Nevertheless, studies in other plant species have demonstrated that PBZ treatment has the opposite effect of GA promoting AR formation either in the presence of exogenous auxin or during exclusive treatment with PBZ [43,61,62,63,64].

In this study, we report that treatment with the GA biosynthesis inhibitor PBZ promotes etiolation-induced AR formation from hypocotyl and wounding-induced AR formation from leaf explants (DNRR) in *Arabidopsis*. Accordingly, we found that GA treatment blocked AR formation in both types of adventitious rooting. This seems to be in conflict with a previous report stating that GA is required for DNRR in *Arabidopsis* leaf explants [50]. By reanalyzing the classical GA biosynthesis and signaling mutants *ga5-1* [65], *ga1-4* [66], and *gai-1* [67] in the Landsberg *erecta* (L*er*-0) background, we found that (i) the receptor-kinase ERECTA (ER) is required for DNRR and (ii) the depletion of GA biosynthesis and signaling promotes DNRR rescuing AR formation that is declined in L*er*-0 leaf explants. We also found a similar reduction in DNRR in *er*-105 mutants in the Col-0 background, which we could rescue using PBZ. Since we found substantial variation in the rate of AR formation in a wild-type background among experiments, we employed *er-105* mutants as genetic tools to find putative effectors of DNRR by whole-genome transcription profiling. Among the differential expressed genes, we found a significant overlap of PBZ-regulated genes with genes directly bound by the DELLA-targets PIF4, BZR1, and/or ARF6 [37] and/or genes regulated by GA [68,69,70] or auxin [68,71]. Among them, we focused on transcription factor genes that are transcriptionally regulated during DNRR and identified several candidate genes, including *LBD16*, a known regulator of DNRR [13]. DNRR assays with an *LBD16::GUS* plant line confirmed the positive response to PBZ treatment, which was abolished by adding GA or blocking polar auxin transport. We discuss how depleted GA signaling can increase *LBD16* expression considering auxin-dependent and auxin-independent pathways, but we also take into account that other candidate genes such as *LATERAL ROOT PRIMORDIUM 1* (*LRP1*) could contribute to the positive effects of PBZ on DNRR. This study revealed that depletion of GA biosynthesis promotes DNRR in *Arabidopsis* and thus will contribute to the further clarification of the role of GA during AR formation in plants. 

## 2. Results

### 2.1. Blocking of GA Synthesis by PBZ Accelerates AR Formation

In *Arabidopsis*, PBZ treatment has not been used during etiolation-induced AR formation from hypocotyl nor wounding-induced DNRR from leaf explants. In contrast, GA is known to inhibit adventitious rooting from hypocotyl in *Arabidopsis* [43]. To test the effects of PBZ on AR formation from *Arabidopsis* hypocotyl, we performed AR formation assays [72] (Figure 1A,B). As expected, treatment with 1 µM GA significantly reduced the AR number in the wild type, while PBZ treatment (5 µM) increased the number of ARs, similar to what was described previously in other plant species [43,62,64]. In line with previous studies [73,74,75,76], we found that PBZ treatment had an overall negative effect on plant growth, resulting in smaller leaves and shorter roots (Figure 1A). To examine the effects of PBZ on the main root system, we grew wild-type plants on media supplemented with different concentrations of PBZ (Appendix A). We found that the main root length decreased proportionally with increased PBZ concentrations, confirming the negative effects of PBZ for main root growth, while the LR number was unaffected. Therefore, only etiolation-induced AR formation from hypocotyl was positively affected by PBZ in *Arabidopsis* wild-type plants. 

Etiolation-induced AR formation from hypocotyl and wounding-induced DNRR from leaf explants share many features, but they partially show opposite responses to hormone treatments such as with jasmonate [1]. Therefore, we also tested the effects of GA and PBZ on DNRR from leaf explants (Figure 1C–E,H). The response pattern was very similar to that of AR formation from hypocotyl; the rooting rate from leaf explants was significantly decreased by GA treatment and increased by treatment with PBZ. Notably, PBZ treatment did not only promote DNRR but also resulted in thicker ARs (Figure 1D,H). This is in line with previous observations in newly developed roots of other plant species [46,77,78,79,80]. 

Next, we tested the effects of PBZ on DNRR in time courses and found that PBZ treatments can accelerate AR formation in leaf explants, reaching rooting rates of 100 percent after 10 DAC (days after culturing started), while untreated plants displayed a rooting rate of about 45 percent (Figure 1F). During DNRR in the dark, auxin biosynthesis by YUC proteins is increased in comparison to light conditions, accelerating the rooting rate in wild-type plants, which can reach rooting rates of almost 100 percent [20]. We therefore performed DNRR assays in darkness and found that AR formation was significantly increased in untreated leaf explants, while additional PBZ treatment can enhance the rooting rates in comparison to the mock-treated plants (Figure 1G). Nevertheless, the accelerating effect of PBZ in the dark on AR formation was not always reflected in the rooting rates but in the increased root length caused by earlier root initiation (Figure 2A,B). 

### 2.2. The Positive Effects of Auxin and PBZ Treatment on DNRR Are Distinguishable 

Previously, it has been suggested that GA inhibits AR formation by perturbing polar auxin transport [43]. Naphthylphthalamic acid (NPA) is an inhibitor of auxin transport by PIN proteins, which can prevent the formation of the auxin maximum and ARs in *Arabidopsis* leaf explants during DNRR [21]. To compare the effects of GA and PBZ treatment with the effects of exogenously applied auxin and NPA on AR formation from *Arabidopsis* leaf explants, we performed DNRR assays under dark conditions (Figure 2). 

Culturing in darkness up-regulates *PIF* genes that trigger dark-induced senescence in *Arabidopsis* leaf explants, which is inhibited in whole darkened plants [81,82]. We observed that GA treatment could accelerate dark-induced senescence in our DNRR assays (Figure 2A). This is in line with GA promoting PIF function by degrading DELLA proteins [83]. Therefore, senescence might play a minor role in reduced AR formation by GA under dark conditions. At 7 DAC, the rooting rate of IAA-treated leaves reached over 20%, while the rooting rate of other treatments remained under 10%. The IAA-driven rooting rate remained significantly stronger than the other treatments and reached over 90% at 11 DAC. In view of the rooting capacity, the enhanced DNRR activity of the IAA-treated leaves was even more eminent (Figure 2C). The PBZ-treated leaf explants had significantly longer roots than the mock-treated leaf explants, which might reflect earlier AR initiation (Figure 2A). However, PBZ had negative effects on the rooting capacity (Figure 2A–C). Therefore, the IAA and PBZ treatments resulted in clearly distinguishable phenotypes. IAA strongly increased the root number, while PBZ promoted an early emergence of a single AR (Figure 2A). This is in line with the assumption that reduced GA signaling by PBZ accelerates auxin transport but does not significantly increase the IAA content that would result in more ARs per leaf explant. In contrast, the AR phenotypes of the NPA and GA treatments were similar but not identical. GA treatment caused significantly decreased rooting rates and capacities, while NPA treatment blocked all DNRR activity (Figure 2B,C). Therefore, GA might decrease auxin transport but does not block the transport entirely.

We nevertheless hypothesized that polar auxin transport is essential for the acceleration of AR formation by PBZ during DNRR. To test this hypothesis, NPA+PBZ treatment was performed. As expect, the rooting rate of the PBZ-treated leaf explants increased faster than in the wild-type explants, while NPA treatment blocked all rooting, which was also the case during NPA+PBZ treatment, indicating that polar auxin transport remains essential for DNRR in the presence of PBZ (Figure 2D,E). We also tested the effects of IAA+PBZ double treatment on DNRR. Although IAA and PBZ treatments resulted in clearly distinguishable rooting phenotypes (Figure 2A), both treatments accelerated the rooting rate from leaf explants (Figure 2F). However, the acceleration of the rooting rate by IAA was stronger and epistatic to the weaker effects of PBZ, while PBZ decreased the rooting capacity even in the presence of IAA (Figure 2F,G). Thus, the IAA+PBZ double treatment did not show additive effects on the rooting rates, which is in line with the hypothesis stating that the accelerating effects caused by PBZ on the rooting rates are mainly based on increased auxin transport and/or signaling. On the other hand, the negative effect of PBZ on the rooting capacity can only partially be compensated by exogenous auxin treatment (Figure 2G).

### 2.3. Loss of GA Biosynthesis Enzymes and GA Signaling Overcome Eradication of AR Formation in Erecta Mutant Leaf Explants 

Recently, it was reported that *Arabidopsis* leaf explants of the classical GA biosynthesis mutant *ga5-1* and GA signaling mutant *gai-1*, which are both in the L*er*-0 background, had lower rooting rates and capacities than the ecotype Col-0 that served as the wild-type control [50]. The *ga5-1* mutant contains a G-to-A point mutation that inserts a translational stop codon in the protein-coding sequence of *GA20ox1*, which encodes an enzyme that catalyzes the final step of GA biosynthesis [65], while the *gai-1* mutant expresses a dominant-negative version of a DELLA protein that is a negative regulator of GA signaling [67,84]. Therefore, both mutants have strongly depleted GA signaling similar to plants treated with PBZ. Nonetheless, the GA mutants and PBZ treatment seemed to have opposite effects on AR formation from *Arabidopsis* leaf explants [50]. In order to clarify these contrary findings, we performed DNRR assays with *gai-1*, *ga5-1*, and another GA biosynthesis mutant *ga requiring 1-4* (*ga1-4*) in the L*er*-0 background [66], while L*er*-0 served as the wild-type control (Figure 3A). Surprisingly, we found that almost all activity of AR formation was eliminated in L*er*-0 leaf explants, suggesting that the ER receptor is a positive regulator of DNRR. In contrast, *ga1-4*, *ga5-1*, and *gai-1* mutants, which all carry the same *er* mutation, displayed stronger DNRR activity than the wild-type L*er*-0. In other words, the loss of GA synthesis and signaling (in *ga1-4*, *ga5-1*, and *gai-1*) can rescue declined AR formation in difficult-to-root leaf explants such as *er* mutants. These findings are in line with the assumption that dampening of GA signaling, either genetically or through PBZ treatment, can increase DNRR activity in *Arabidopsis* leaf explants.

### 2.4. PBZ Treatment Can Rescue Root Regeneration in ER Mutants 

Since we found that almost all rooting activity was abolished in L*er*-0 leaf explants, we hypothesized that *ER* signaling is a positive regulator of DNRR. To confirm this hypothesis, we employed a second *er* allele in our DNRR assays. *ER* encodes a member of the receptor-like protein kinase (RLK) gene family, containing an extracellular receptor domain with 20 leucine-rich repeats (LRRs), a transmembrane domain, and a cytoplasmic serine/threonine protein kinase domain [85]. The importance of the protein kinase domain for *ER* function is illustrated by the L*er*-0 allele that carries a missense mutation of isoleucine 750 to lysine, which is a conserved residue in most RLKs [86]. This allele is equivalent in phenotypic strength to *er-105* that has no mRNA signal on a Northern blot [85]. Thus, both *er* alleles, *er-1* (L*er*-0) and *er-105* (Col-0), are likely to be null alleles [85,86]. In DNRR time courses, we found no significant differences between L*er*-0 and *er-105*, while both *er* alleles displayed significantly reduced rooting rates compared to Col-0 (Figure 3B).

Since mutants with a loss of GA synthesis or GA signaling can overcome the depletion of AR formation in L*er*-0 leaf explants (Figure 2A), we assumed that PBZ treatment can rescue the decreased rooting rate in both *er* alleles. To test this assumption, we performed DNRR assays with both *er* alleles and Col-0 (Figure 3C). PBZ treatment increased the rooting rate in all three lines. Although both *er* mutants did not reach the same rooting rate like the wild-type Col-0 after PBZ treatment, both *er* alleles displayed a much higher ratio of PBZ to mock-treated leaf explants than the wild type. We conclude that this significantly higher difference in rooting between the PBZ treatment and the mock treatment in *er* mutants could be a big advantage in whole-genome transcription profiling during DNRR. 

### 2.5. Transcriptomic Analysis, Using ER Leaf Explants, Revealed That PBZ Treatment Suppresses GA and Brassinosteroids Responses While Inducing Expression of the Rooting Factor LBD16 

To identify genes that regulate DNRR and are differentially expressed by PBZ treatment, we performed whole-genome transcription profiling. Under LD conditions, the average rooting rates of wild-type (Col-0) leaf explants strongly vary in independent experiments, ranging from 30 to 100 percent [26,50], while under dark conditions, the rooting-promoting effects of PBZ were occasionally covered, and the mock- and PBZ-treated Col-0 leaf explants could reach similar high rooting rates (Figure 1G and Figure 2B). In contrast, the rooting rate ratios of PBZ-treated leaves to mock-treated leaves were significantly higher and less variable in the *er* mutants compared with Col-0 (Figure 3C). We therefore employed the *er-105* mutant as a genetic tool in our whole-genome expression analysis using RNA seq and cultured *er-105* leaf explants in darkness with and without PBZ for 24 h (24 HAC; hours after cultivation started). Comparing PBZ-treated to mock-treated *er-105* mutants, genes with a ≥1.5-fold expression change and a *p*-value ≤ 0.05 were considered to have significantly different expression levels. Differentially expressed genes (DEGs) are listed in Appendix A. A total of 772 genes were regulated by PBZ (336 up-regulated and 436 down-regulated) in leaf explants 24 HAC (Figure 4A). This corresponds to 2.8% of the 27,655 protein-coding loci in the *Arabidopsis* genome [87]. 

In order to verify a decreased GA response by PBZ, we compared our RNA seq data with GA-regulated genes identified in previous studies [68,69,70]. As expected, we found a significant overlap of PBZ-repressed genes with GA-induced genes (*p* < 3.9 × 10^−9^) and PBZ-induced genes with GA-repressed genes (*p* < 1.2 × 10^−6^), indicating a reduced GA response in the PBZ-treated *er-105* leaf explants (Appendix A). These findings are in line with the assumption that PBZ disrupted GA biosynthesis, which ultimately results in reduced GA signaling and GA response. This conclusion was supported by the increased expression of the GA synthesis genes *GA3ox1/GA4*, *GA20ox1/GA5*, *GA20ox2,* and *GA20ox3* (Figure 4E and Appendix A), which act as negative feedback regulated by GA [88,89]. 

The hormone signaling pathways of GA and brassinosteroids (BRs) are directly linked through the DELLA-PIF4-BZR1 module [90]. We therefore compared the PBZ-regulated genes with BR-regulated genes identified in previous studies [68,91]. As expected, we found evidence for reduced BR responses with significant overlaps of PBZ-repressed genes with BR-induced genes (*p* < 4.7 × 10^−8^) and PBZ-induced genes with BR-repressed genes (*p* < 1.6 × 10^−3^; Appendix A). Notably, five genes, *DWF4/CYP90B1*, *CPD/CYP90A1*, *CYP90D1*, *ROT3/CYP90C1*, and *BR6OX2/CYP85A2*, encoding enzymes of the BR biosynthesis pathway [92], were up-regulated by PBZ treatment (Figure 4E and Appendix A). All of these BR biosynthesis genes are direct targets of BZR1 [91]. More than 60% of the PBZ-repressed genes and more than 40% of the PBZ-induced genes are known to also be regulated by GA and/or BR (Figure 4B). This suggests that approximately half of the transcriptional changes caused by PBZ were due to direct interference with GA and/or BR signaling.

Several review articles promote the idea that PBZ treatment not only blocks GA biosynthesis but also ABA catabolism [93,94,95,96]. Most recently, Kang et al., 2023, measured the ABA content after PBZ treatment in Chinese cabbage (*Brassica rapa* ssp. *Pekinensis*). Although they found increased endogenous ABA concentrations under normal conditions, drought stress significantly increased the ABA levels, whereas PBZ inoculations significantly decreased them [97]. Similarly, in *Phoebe bournei* roots, PBZ treatment reduced the ABA content [98]. Therefore, it seems less evident than the reviews implying whether PBZ treatment increases the ABA content in general. Depending on the tissue and other factors, such as drought stress conditions that likely occur in detached leaves during DNRR, PBZ can decrease the ABA content. Furthermore, in many research articles, the ABA content is rarely measured, and the assumed higher ABA levels obtained by PBZ are often concluded using transcriptional profiling. This can be problematic since GA and ABA signaling antagonize each other’s responses [99], and it is hard to distinguish transcriptional responses to lower GA from responses to a higher ABA content. Nevertheless, we compared the PBZ-regulated genes with ABA-regulated genes identified in a previous study [68]. We found a significant overlap of PBZ-induced genes with ABA-induced genes (*p* < 2.4 × 10^−14^) and PBZ-repressed genes with ABA-repressed genes (*p* < 9.7 × 10^−4^), but 74% (*p* < 2.9 × 10^−7^) to 88% (*p* < 2.1 × 10^−12^) of these genes are also known to be conversely regulated by GA/BR signaling (Appendix A). Next, we showed that even if PBZ would increase the ABA content, ABA treatment inhibits AR formation, even in the presence of endogenous auxin (Appendix A), which is not in line with our observation that PBZ promotes AR formation, and from then onwards, we ignored ABA as an agent in our DNRR study. 

To narrow down which of the PBZ-regulated genes are likely involved in AR formation, we compared them with wounding-regulated genes in leaf explants during DNRR [26]. We found a large overlap of genes that were regulated by PBZ and wounding (Appendix A). The overlap of genes up-regulated by PBZ and wounding was most significant (71 genes *p* < 1.1 × 10^−19^), followed by the overlap of genes down-regulated by PBZ and wounding (44 genes *p* < 3.0 × 10^−9^), and the overlap of genes up-regulated by wounding but down-regulated by PBZ (47 genes *p* < 4.1 × 10^−4^). A total of 174 genes were regulated by both PBZ and wounding, accounting for 25% of the 694 PBZ-regulated genes. This finding suggests that a large number of the PBZ-regulated genes are specifically involved in DNRR.

Since auxin signaling is essential for AR formation during DNRR [20,21,100], we compared our RNA seq data with IAA-regulated genes identified in previous studies [68,71]. Although we found large overlaps between PBZ- and auxin-regulated genes, their expression changes correlated positively or negatively (Appendix A). However, genes regulated by PBZ- and auxin largely overlapped with genes regulated by PBZ and GA/BR (Figure 4B). Next, we identified PBZ-regulated genes known to participate in auxin metabolism, transport, and signal transduction (Figure 4C,D and Appendix A). Among the genes up-regulated by PBZ treatment, we found two IAA-synthesis genes, *NIT2* [101] and *YUC9* [20]; the MeIAA hydrolase gene, *METHYL ESTERASE 16* (*MES16*) [102]; and the UDP-glucosyltransferase *UGT74E2* that indirectly increases the IBA content [103]. Together, they can increase the content of biological active auxins. Likely in response to increased auxin levels, we found three *GRETCHEN HAGEN 3* (*GH3*) genes, *GH3.5/WES1*, *GH3.15*, and *GH3.17*, which were up-regulated by PBZ. These GH3 genes encode IAA/IBA-amido synthetases that can decrease the amount of free auxin [104,105,106]. Although the PBZ treatment resulted in expression changes in auxin metabolism genes, they are likely not related to expression changes in the jasmonate-SPL10/11-ERF109/111-JAZ module [25,26] since *ABR1/ERF111*, *SPL10*, *SPL11*, and especially the auxin biosynthesis gene *ASA1*, its key target, were not significantly up-regulated (Appendix A), although the expression of *ERF109* was increased and *JAZ8* expression was abolished by PBZ (Figure 4D and Appendix A). We found the intracellular auxin transporter genes *PILS5* [107] and *PIN1*, encoding an essential auxin efflux carrier [108] induced by PBZ (Figure 4C–E and Appendix A, while negative regulators of the polar auxin transport by PIN1 were repressed (Appendix A). The increased expression of *MAP KINASE KINASE7* (*MKK7*) causes, via MPK6, phosphorylation of the PIN1 S337 site, which disturbs PIN1 polarity and auxin gradients [109,110]. Thus, reduced *MKK7* expression by PBZ treatment should lead to stable auxin transport. Similarly, *PATL5*, another significantly down-regulated gene by PBZ, is also involved in the PIN1 location at the cell membrane by repolarization [111]. Taken together, our transcriptional profiling results suggest that PBZ at least partially promotes DNRR by increased auxin biosynthesis and transport and, ultimately, auxin signaling and response. 

Therefore, we extended our search for potential positive regulators of DNRR by PBZ to genes that are not GA/BR response genes but inducible by wounding and auxin (Figure 4E). Focusing on transcription factors, we identified *LBD16* as a gene that is inducible by auxin treatment and wounding (during DNRR) [13], while its promoter can be bound by the DELLA-targets PIF4, BZR1, and/or ARF6 [37]. Remarkably, *HOMEOBOX PROTEIN* (*HB23*) encoding a zinc finger homeobox gene and the known transcriptional repressor of *LBD16* [112] was down-regulated, which could contribute to the higher *LBD16* expression obtained by PBZ treatment (Figure 4C–E).

To confirm that PBZ is a positive regulator of *LBD16* expression, and consequently, that GA represses *LBD16*, we employed *LBD16::GUS* plants [113] in DNRR assays (Figure 5). As expected, PBZ treatment increased *LBD16::GUS* expression, while additional GA treatment significantly decreased *LBD16::GUS* expression, and additional NPA treatment abolished all expression. These results suggest that *LBD16*, a known positive regulator of DNRR [13,114], is directly or indirectly repressed by GA signaling, which can be overcome by PBZ treatment that functions at least partially through changes in auxin transport and/or auxin signaling. Nevertheless, we also found other candidate genes such as *LRP1*, which could encode GA/BR-related DNRR factors working beyond and in parallel to *LBD16* (Figure 4C–E).

## 3. Discussion

In this study, we examined the effects of GA and its biosynthesis inhibitor PBZ on wounding-induced AR formation from *Arabidopsis* leaf explants (DNRR). Our interest in GA signaling and adventitious rooting was triggered by two contradictory studies in *Arabidopsis* stating either that GA inhibits etiolation-induced AR formation from hypocotyl [43] or GA is required to promote DNRR through the tight regulation of GA signaling through DELLA repressors [50]. We wondered whether GA could play reverse roles in both types of AR formation since jasmonate has such contrary effects [1,25,28]. Since the effects of GA and PBZ treatment on AR formation in *Arabidopsis* leaf explants were not studied yet, we conducted these DNRR assays and compared the results with the GA and PBZ responses of etiolation-induced AR formation in hypocotyl. Under our conditions, GA treatment inhibited AR formation in both types of AR, while PBZ treatment resulted in accelerating effects on adventitious rooting. These findings are in line with the large canon of studies in other plant species [41,42,43,44,45,46,47,48,61,62,63,64]. Accordingly, we found that the loss of GA synthesis and signaling in classical GA mutants promotes DNRR. The discrepancy of our results with [50] and the novel role of *ER* signaling in DNRR might be clarified in future studies. Nevertheless, our study suggests that PBZ treatment has a direct effect on GA and BR signaling during DNRR. Transcriptional profiling at 24 HAC can only provide a glimpse of the molecular mechanisms of PBZ-controlled AR formation in leaf explants. Further studies of early events and genetic validation will be necessary to obtain a full picture. Nevertheless, our transcription profiling process and the literature provide enough information for conceptional models (Figure 4C and Appendix A). 

### 3.1. Probable Direct Effects of PBZ Treatment on GA and BR Signaling During DNRR

This study revealed that GA inhibits DNRR, while the GA synthesis inhibitor PBZ promotes AR but not LR formation in *Arabidopsis* (Figure 1 and Appendix A). Our hormone and hormone inhibitor DNRR assays as well as expression data provided a brief glimpse into the putative GA pathways during DNRR. PBZ treatment in darkness results in the accumulation of DELLA proteins that block GA signaling [58,115]. The hormone signaling pathways of GA and brassinosteroids (BRs) are directly linked through the DELLA-PIF4-BZR1 module [90]. The BR signaling cascade activates the transcription factor BZR1 that directly regulates BR-responsive gene expression and development [116]. BZR1 and PIF4 form heterodimers that directly bind and synergistically activate common target genes, which is inhibited by DELLA proteins. Nevertheless, BZR1 and PIF4 have partially opposite effects on the expression of common target genes. In our conceptional model, the accumulation of DELLAs by PBZ treatment blocks BZR1/PIF4-dependent GA and BR signaling in leaf explants (Figure 4C). Reduced BZR1 function by DELLA could directly result in the up-regulation of the auxin biosynthesis genes *NIT2* and *MES16* and the down-regulation of *IAA6/SHY*, *IAA14/SLR*, and *HB23*, which can promote *LBD16* expression.

A total of 396 genes (51.3%) of the 772 PBZ-regulated genes are bound by the DELLA targets PIF4, BZR1, and/or ARF6 (Appendix A) [37]. This is in line with the hypothesis that GA signaling directly controls DNRR. On the other hand, the huge number of PIF4, BZR1, and/or ARF6 targets among the PBZ-regulated genes makes it impossible to investigate the relevance of all of them for PBZ-promoted DNRR. Many GA and BR biosynthesis genes are feedback-regulated. Accordingly, our expression data confirm that the PBZ treatment dampened GA and BR responses indicated by the increased expression of the GA biosynthesis genes *GA3ox1*, *GA20ox2*, and *GA20ox3* (Figure 4E and Appendix A), which are negatively feedback-regulated by GA [88,89]. The BR biosynthesis genes *DWF4/CYP90B1*, *CPD/CYP90A1*, *CYP90D1*, *ROT3/CYP90C1*, and *BR6OX2/CYP85A2*, which are direct targets of BZR1 [91,92], were also up-regulated by PBZ treatment (Figure 4E and Appendix A). Furthermore, we compared the expression of *GAI*, *RGA1*, and *INDOLE-3-ACETIC ACID INDUCIBLE 7/AUXIN RESISTANT 2* (*IAA7/AXR2*), which are well-known marker genes of GA response [69] in *er-105* mutant leaf explants with and without PBZ treatment in comparison with Col-0 (Appendix A). As expected, *GAI*, *RGA1*, and *IAA7/AXR2* were down-regulated by PBZ.

The down-regulation of several bHLH transcription factor genes, such as *IBL1*, *PRE1*, *HBL1*, *BEE2*, and *PAR1*, could be a direct consequence of reduced BZR1 and PIF function due to DELLA accumulation by PBZ, which could be enhanced by the up-regulation of their repressor IBH1 (Appendix A). BZR1, PIF4, and/or ARF6 interact and activate *PRE1*, which is inhibited by DELLA [37]. The full name of *PRE1*, *PACLOBUTRAZOL RESISTANCE 1*, indicates the importance of this gene for PBZ response and it is therefore a good candidate for DNRR studies. Furthermore, the PBZ-repressed transcription factor genes *GRXS13*, *bZIP61*, *bZIP34*, and *GASA4* and the auxin receptor gene *GRH1* are direct targets of BZR1 (Appendix A) and therefore can be interesting for future DNRR studies, although all five genes could be down-regulated by increased ABA (Appendix A). In conclusion, many of the PBZ-regulated transcription factor genes are direct targets of PIF4, BZR1, and/or ARF6, suggesting a direct role of PBZ and, consequently, GA signaling in AR formation during DNRR.

### 3.2. Probable Direct and Indirect Effects of PBZ Treatment on Expression of Rooting Factor LBD16 During DNRR

There are at least four probable pathways of how the depletion of GA signaling by PBZ can increase the expression of *LBD16* (Figure 4C): (i) A higher expression of auxin biosynthesis genes, such as *NIT2* and *YUC9*, and auxin transport genes, such as *PIN1*, may increase the auxin response that should also be enhanced by (ii) the down-regulation of *IAA6/SHY* and *IAA14/SLR* (Figure 4D) but also other Aux/IAAs such as *IAA7/AXR2* (Appendix A). (iii) The down-regulation of *HB23*, encoding a repressor of *LBD16* [112], should have a more direct impact of *LBD16* expression. Since the promoter of *LBD16* is directly bound by PIF4 and BZR1, we could speculate that (iv) they are repressors of *LBD16*, and their reduced function through DELLA accumulation would be the most direct mechanism of regulating *LBD16* expression by PBZ. The precise contribution of these pathways on the transcriptional up-regulation of *LBD16* might be the subject in future GA/DNRR studies. 

### 3.3. Probable GA/BR-Related DNRR Factors Beyond and in Parallel to LBD16 

Transcription factor genes, which are transcriptionally regulated by wounding signaling during DNRR [26], are naturally good candidates for being regulators of AR formation in leaf explants. Our expression analysis revealed that seven of the wounding-repressed transcription factor genes [26], which are *IAA14/SLR*, *HB23*, *HB21*, *GRXS13*, *IBL1*, *PAR1*, and *GASA4*, were stronger down-regulated by PBZ treatment. On the other hand, the wounding-activated transcription factor genes [26], which are *LBD16*, *SPEEDY HYPONASTIC GROWTH* (*SHG/NAC47*), *DEHYDRATION RESPONSE ELEMENT-BINDING PROTEIN 19* (*DREB19*), *OVATE FAMILY PROTEIN 1* (*OFP1*), and *LRP1*, were more strongly up-regulated by PBZ in our DNRR expression analysis (Figure 4E). The correlation of up-regulation by wounding signaling [26] and even stronger up-regulation by PBZ treatment (this study) is a strong indication that these transcription factors are (i) positive regulators of DNRR and that (ii) they are negatively regulated by GA signaling and vice versa. Notably, only the PBZ-induced down-regulation of *IDD16* (Figure 4E and Appendix A), encoding a transcription factor that promotes auxin biosynthesis [117], broke the pattern since *IDD16* is up-regulated by wounding signaling [26]. 

Like *HB23*, *HB21* and its paralogue *HB31* were down-regulated in our expression data (Appendix A). *HB21* controls shoot branching, while HB21 together with *HB31* control floral architecture through the miR396/GRF module [118]. Notably, the *miR396*/GROWTH-REGULATING FACTOR (GRF) module is known to form a negative feedback loop with *PLETHORA* (*PLT*) genes in the RAM [119]: The GRFs repress *PLTs*, regulating their spatial expression gradient. On the contrary, PLTs activate *MIR396* in the root stem cells to repress the *GRFs*. Thus, *HB21* and *HB31* are involved in shoot and root development and could play a role in AR formation via the miR396/GRF module. *GRXS13*, encoding a member of the CC-type glutaredoxin (ROXY) family, can interact with the transcription factor TGA2 and suppress *ORA59* promoter activity that can be induced by jasmonate signaling [120]. However, *ORA59* is down-regulated as well (Appendix A), and therefore, the relevance of repressing *GRXS13* by PBZ for DNRR remains elusive. *GASA4* promotes GA responses [121], and therefore, the down-regulation of *GASA4* by PBZ should promote DNRR. The bHLH transcription factor genes *IBL1* and *PAR1* are direct targets of PIF4 and BZR1, and *PAR1* is also a target of IBL1, while IBL1 antagonizes PIF4 but not the PIF4–BZR1 dimer [90]. Their deep interconnection with the GA signaling pathway and down-regulation during DNRR [26] make *IBL1* and *PAR1* good candidates for future DNRR studies. 

*SHG/NAC47* promotes hyponastic growth that involves the volatile phytohormone ethylene [122,123] that might switch the AR-inhibiting effect of GA into an AR-promoting pathway as observed in rice [49], but whether *SHG/NAC47* promotes AR formation is unclear. *DREB19* is involved in the response to drought stress [124] that naturally appears in detached leaves, but it also remains unclear whether *DREB19* promotes AR formation. OFP1 is a member of the plant-specific ovate protein family that can bind to KNOX and BELL-like TALE class homeodomain proteins to relocate these TALE homeodomains from the nucleus to the cytoplasm. Thus, OFP1 functions as a transcriptional repressor, while one of its known target genes is *GA20ox1/GA5* [125]. From there, OFP1 might suppress GA biosynthesis through the repression of *GA20ox1/GA5* during DNRR to promote AR formation. Nevertheless, *LRP1* is, besides *LBD16*, the most probable candidate to significantly affect GA-regulated DNRR since *LRP1* is an early wounding response gene during DNRR [26], responsible to auxin but not ABA, and is directly bound by BZR1 and ARF6 [37]. Furthermore, *LRP1* is expressed in LR and AR primordia [126], and the overexpression of *LRP1* results in reduced emerged LR density, which might be related to the reduced rooting capacity by PBZ treatment (Figure 2). Although *LRP1* expression is induced by auxin during AR and LR development [126,127], it belongs to neither the fast IAA response genes in tissue culture [71] nor in seedlings [68]. Differential sensitivities of developmental responses to auxin are well known. Unlike *arf7 arf19* double mutants, *LRP1* OE plants showed normal auxin sensitivity. In future studies, it would be interesting to explore how *LRP1*, *OFP1*, and the other candidates contribute to AR formation during DNRR. 

## 4. Materials and Methods

### 4.1. Plant Materials and Culture Conditions

Most experiments were performed with the *Arabidopsis* ecotype Columbia (Col-0). The receptor mutant *er-105* [85,86] and *LBD16::GUS* plants [113], both in a Col-0 background, were also used. Landsberg *erecta* (L*er*-0) served as the wild-type control for the GA mutants *gai-1* [67], *ga1-4* [66], and *ga5-1* [65]. After surface sterilization with 70% ethanol (twice), the seeds were sowed on half-strength Murashige and Skoog (½ MS) plates with 10 g/L of sucrose, 0.5 g/L of MES, and 9 g/L agar with pH = 5.8 and stratified at 4 °C for 3 days. Then, the plates were transferred to a plant growth cabinet with long-day (LD) conditions (16/8 h light/dark cycle) at 22 °C. 

### 4.2. De Novo Root Regeneration (DNRR) and Hormone Treatment

The first pair of rosette leaves (primary leaves) from 14-day-old *Arabidopsis* plants were used for DNRR assays. The growth conditions of plants and culture conditions of leaf explants were based on the protocols described earlier [128]. The leaf explants were cultured on B5 medium (Gamborg B5 medium, Phytotech Lab, Lenexa, KS, USA, with 0.5 g/L MES and 8g/L Agar, pH = 5.7) with 30 g/L of sucrose under dark conditions or LD without sucrose. Wild-type (Col-0) plants were germinated by incubating the seeds in light for 3–5 h before being grown in the dark for 3 days to induce hypocotyl elongation. Well-elongated and etiolated seedlings were transferred to various treatments under long-day (LD) light conditions for 7 days, and then, the number of ARs were determined. The rooting rate is represented by the percentage of explants with regenerated roots at a given time point [128], while the rooting capacity is represented by the percentage of leaf explants with different numbers of regenerated roots [26]. For each experiment, more than six plates of each treatment with more than 10 explants per genotype were used. For hormone and hormone inhibitor treatments, we used 0.1 µM IAA (Sigma-Aldrich, Inc. St. Louis, MO, USA), 1 µM NPA (Sigma-Aldrich, Inc. St. Louis, MO, USA), 0.1 µM ABA (Sigma-Aldrich, Inc. St. Louis, MO, USA), 1 µM GA3 (Phytotech Lab, Lenexa, KS, USA), and/or 5 µM of paclobutrazol (PBZ) (Solarbio, LIFE SCIENCES, Beijing, China/Phytotech Lab, Lenexa, KS, USA) in the B5 medium. A one-tailed Student’s *t* test, resulting in *p* values, was employed to assess the statistical significance between value pairs of different treatments [129]. For RNA seq materials, leaf explants were cultured on B5 medium plates with and without 5 µM of PBZ for 24 h, harvested in liquid nitrogen, and stored in −80 °C.

### 4.3. Adventitious Root Formation from Etiolated Hypocotyls

Col-0 plants were grown in vitro. The AR induction conditions were previously described [72]. After sterilization, the seeds were sown in lines on plates and left for 3 days at 4 °C for stratification, exposed to light for 3 h to induce germination, transferred to the dark for 3 days (until hypocotyls were around 6 mm in length), and then exposed to LD conditions in a plant growth cabinet for 7 days before counting the ARs formed on the hypocotyls. To apply the treatment with GA and PBZ, the etiolated seedlings were carefully transferred to new plates of ½ MS medium with or without supplementary 1.0 µM GA3 or 1.0 µM PBZ and then exposed to light for 7 days.

### 4.4. Root Length and Lateral Root (LR) Number Assays

Root growth assays were performed with minor modifications as described before [130]. In brief, the primary root length was measured under different concentrations of PBZ. Seeds were sowed on ½ MS plates, kept at 4 °C for 3 days, and then incubated in a plant growth chamber at 22 °C for 5 days. Then, the plants were transferred to the ½ MS with 1 µM, 5 µM, and 25 µM PBZ for 7 days, and the length was measured using Image J: a reference with a known length was employed to calibrate using the ‘Set Scale’ function. After calibration, all measurement results are automatically converted. After using the ‘Freehand Line’ tool to draw a line along the main root, the ‘Measure’ function was used, and the measurement results (here, the root length values) are displayed in the ‘Results’ window’. The number of LRs was counted by mature LRs greater than 0.5 mm in length. At least 23 plants were used for each treatment.

### 4.5. RNA Seq and Expression Analysis

Total RNA was isolated and used for the sample preparations. After RNA extraction, mRNA was purified from total RNA by poly-T oligo-attached magnetic beads and cut into small fragments of preferentially 370–420 bp in size for cDNA synthesis. The libraries were purified using the AMPure XP system (Beverly, MA, USA). The qualified libraries were pooled and sequenced on IIIumina platforms. We used fastqc to evaluate the raw data and trim-galore to filter low-quality bases, N bases, and adapters to obtain clean data for further analysis [131]. In order to obtain gene quantitative information, we used hisat2 software (https://github.com/DaehwanKimLab/hisat2.git accessed on 14 October 2024) to align the clean data to the reference genome [132]. Then, StringTie software (https://ccb.jhu.edu/software/stringtie/ accessed on 14 October 2024) was used for the gene transcript quantification analysis. The R package DESeq2 was used to perform an analysis of differentially expressed genes [133]. Further analyses and comparisons were conducted for differentially expressed genes (DEGs, with a ≥1.5-fold expression change and a *p*-value *p* ≤ 0.05), expression fold changes were calculated with a regular spreadsheet program (Excel, Microsoft), and the DEGs are listed in Appendix A with normalized counts per million (CPM), ±standard error of the mean (±SEM). In addition, significantly different expressions were verified using Student’s *t*-test. Venny 2.1 (https://bioinfogp.cnb.csic.es/tools/venny/index.html accessed on 14 October 2024) was used for the comparisons of gene lists: the PBZ-regulated genes identified in this study were compared with published lists of DEG in response to hormones, such as GA [68,69,70], BR [68,91], ABA [68] and auxin [68,71] and wounding-regulated genes during DNRR [26]. 

The detection of β-glucuronidase (GUS) activity in leaf explants was performed as described with minor modifications [134]. In brief, leaf explants were harvested at specific time points (0 HAC, 24 HAC, and 48 HAC) and immersed in the GUS assay solution (50 mM NaHPO4, 0.5 mM ferrocyanide, 0.5 mM ferricyanide, and 1% Triton X-100, pH 7.2) containing 1 mM X-Gluc. The leaves in the GUS solution were vacuum-infiltrated for 30 min and then incubated at 37 °C overnight. To remove the chlorophyll, stained leaves were carried through ethanol series and then photographed with a stereomicroscope. The digital photographs were collated with Adobe Photoshop and adjusted as described before [135].

## Figures and Tables

**Figure 1 ijms-25-13340-f001:**
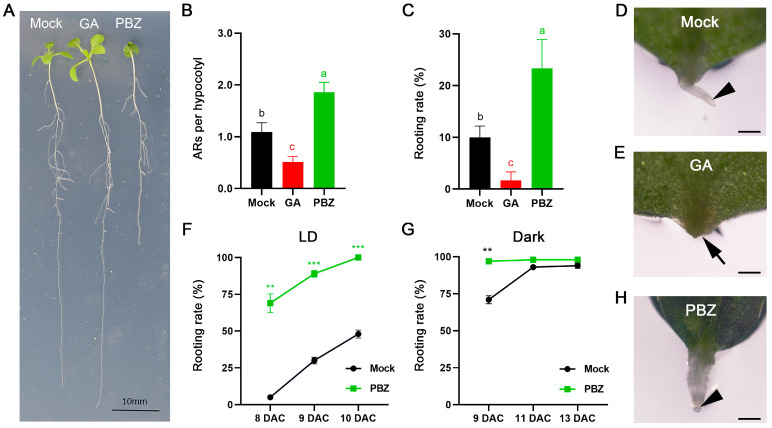
Effects of GA and PBZ treatments on rooting rate of AR formation from hypocotyls and DNRR. (**A**,**B**) AR formation from hypocotyls. (**A**) Rooting phenotype of Col-0 seedlings under mock, GA, and PBZ treatments. Scale bar = 10 mm. (**B**) ARs per hypocotyls (Col-0) under mock, GA, and PBZ treatments 7 days after shift from dark to LD, 10 DAG. (**C**–**H**) DNRR from Col-0 explants under mock, GA, and PBZ treatments (LD and dark conditions). (**C**) Rooting rate of Col-0 leaf explants under mock, GA, and PBZ treatments 9 DAC (LD). (**D**,**E**,**H**) Phenotype of Col-0 leaf explants under mock, GA, and PBZ treatments 9 DAC (LD). Arrow indicates the absence of an AR (**E**). Note that roots are thicker after PBZ treatment in comparison to mock treatment (arrow heads in (**D**,**H**)). Photos were taken using Nikon (SMZ25) microscope. Scale bar = 500 µm. (**G**) Dark conditions accelerate rooting additive to PBZ treatment. (**B**,**C**,**F**,**G**) Average values are shown, ± SEM. Color letters ((**B**,**C**); One way ANOVA, *p* < 0.05) and asterisks ((**F**,**G**); Student’s *t*-test: ** *p* < 0.01; *** *p* < 0.001) indicate significant changes.

**Figure 2 ijms-25-13340-f002:**
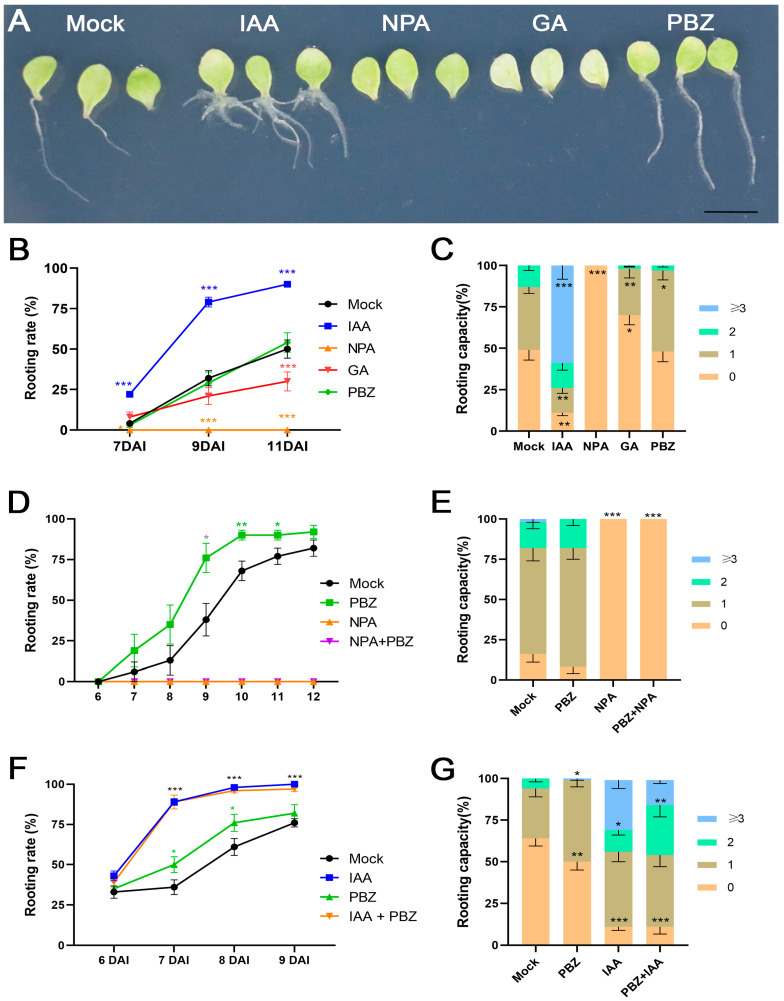
Effects of different hormone treatments on rooting rate (time courses) and rooting capacity in wild-type (Col-0) leaf explants in darkness. (**A**–**C**) Single treatments with IAA, NPA, GA, and PBZ. (**A**) Rooting phenotype of Col-0 leaf explants. Scale bar = 10 mm. (**B**) Rooting rate. (**C**) Rooting capacity. Note that rooting capacity was decreased by PBZ, while IAA treatment increased rooting rate and rooting capacity. (**D**,**E**) Treatments with NPA, PBZ, and NPA+PBZ. Note that rooting is completely abolished by NPA treatment even in presence of PBZ, indicating that polar auxin transport is essential for accelerating effects of PBZ on DNRR. (**F**,**G**) Treatments with IAA, PBZ, and IAA + PBZ. Note that accelerating effects on rooting rates by IAA is epistatic to PBZ treatment (no additive effect), but PBZ significantly reduced rooting capacity even in presence of IAA. Treatments: 0.1 µM IAA, 1 µM NPA, 1 µM GA, and 5 µM PBZ. (**B**–**G**) Average values are shown, ± SEM. Asterisks indicate significant changes (Student’s *t*-test: * *p* < 0.05; ** *p* < 0.01; *** *p* < 0.001).

**Figure 3 ijms-25-13340-f003:**
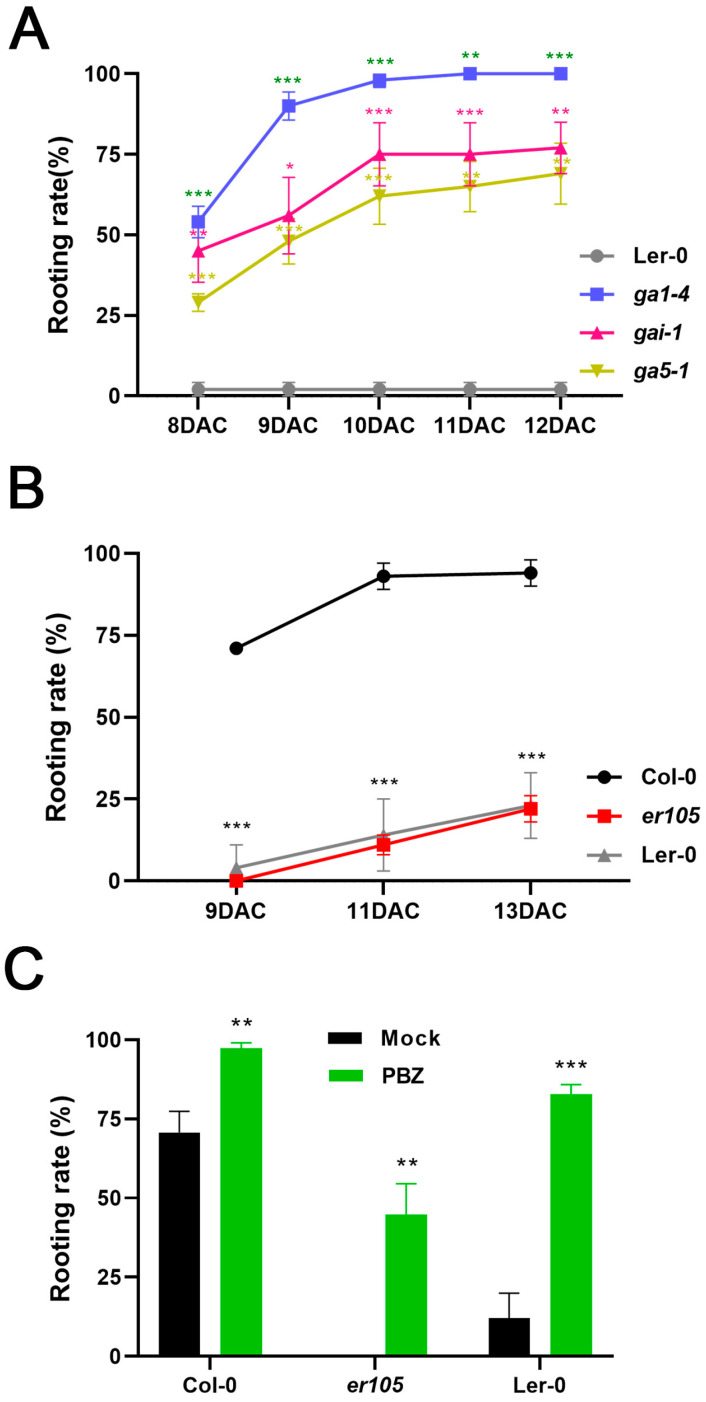
Reanalysis of GA biosynthesis and GA signaling mutants reveals (i) negative effects of GA and (ii) positive effects of ER signaling on DNRR in darkness. (**A**) Rooting rate time course: rooting rate of three GA mutants, *ga1-4*, *ga5-1*, and *gai-1*, are significantly higher than those in wild type (L*er*-0). (**B**) Rooting rate time course: in comparison to wild-type Col-0, rooting was almost abolished in leaf explants of both *er-1* (L*er*-0) and *er-105* (Col-0) mutants. (**C**) Treatment with 5 µM PBZ can recue low rooting rates of both *er-1* (L*er*-0) and *er-105* (Col-0) mutants (9 DAC). (**A**–**C**) Average values are shown, ± SEM. Asterisks indicate significant changes (Student’s *t*-test: * *p* < 0.05; ** *p* < 0.01; *** *p* < 0.001).

**Figure 4 ijms-25-13340-f004:**
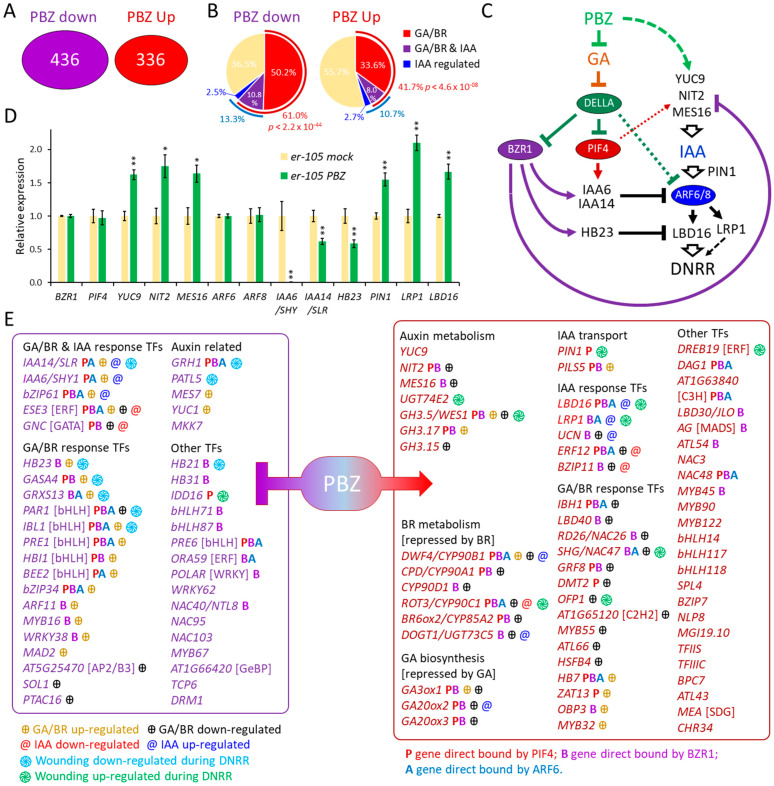
Expression changes in response to PBZ treatment 24 HAC. (**A**) Number of genes with significantly decreased (down) and increased (up) expression rates. (**B**) Percentage of GA/BR- and auxin-regulated genes and their overlaps. (**C**) Conceptional model of potential pathways explaining how PBZ could increase *LBD16* expression. Dotted lines represent known interactions that seem to be covered during DNRR with PBZ. Broken lines represent putative, probable indirect interactions. Note that most arrows indicate transcriptional regulation, but DELLA proteins bind directly to BZR1, PIF, and ARF6 proteins, PBZ blocks GA synthesis, and PIN1 is auxin (IAA) transporter. (**D**) Expression changes in response to PBZ treatment (24 HAC) of genes that can theoretically increase *LBD16* and *LRP1* expression levels. Relative expression in PBZ-treated *er-105* leaf explants compared to mock treatment using CPM values in Appendix A; ±SEM; n = 4. Asterisks indicate significant changes (Student’s *t*-test: * *p* < 0.05; ** *p* < 0.01). (**E**) Selection of genes with significantly decreased (down-regulated) and increased (up-regulated) expression rates. Regulation by GA/BR, IAA, and/or wounding as well as promoter binding by PIF4, BZR1, and/or ARF6 is indicated.

**Figure 5 ijms-25-13340-f005:**
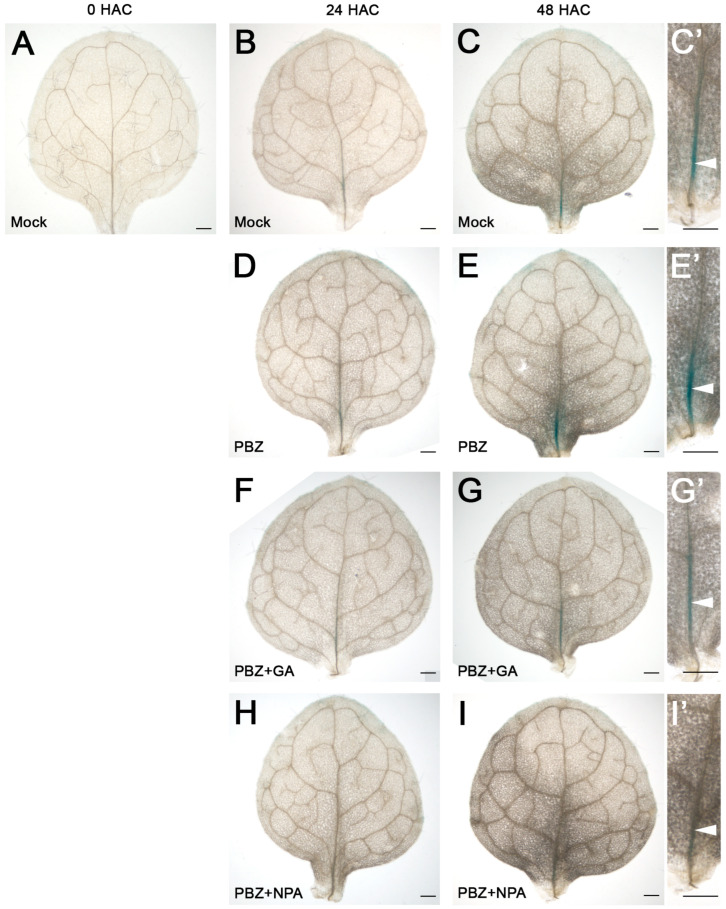
Time courses of LBD16::GUS expression in response to mock, PBZ, PBZ + GA, and PBZ + NPA treatments. HAC, hours after cultivation (induction). (**A**–**C**) Mock treatment. Note that *LBD16::GUS* expression is evidently visible at future rooting site after 48 HAC (**C**,**C**’). (**D**,**E**) PBZ treatment. Note that *LBD16::GUS* expression is stronger after PBZ treatment (48 HAC, **E**,**E**’). (**D**,**E**) PBZ+GA treatment. Note that *LBD16::GUS* expression is strongly reduced by GA (48 HAC, **G**,**G**’). (**F**,**G**) PBZ + GA treatment. Note that *LBD16::GUS* expression is strongly reduced by GA (48 HAC, **G**,**G**’). (**H**,**I**) PBZ + NPA treatment. Note that LBD16::GUS expression is fully quenched by NPA (48 HAC, **I**,**I**’). White arrow heads indicate the *LBD16::GUS* staining, its changes (**C**’,**E**’,**G**’) and/or its absence (**I**’). Photos were taken using Olympus (CX43) microscope. Scale bar = 250 µm.

## Data Availability

The RNA seq data were deposited in BioProject under the ID PRJNA1176316. All other relevant data are included within this article and the Appendix A online.

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
