# Peer review of "Depletion of Gibberellin Signaling Up-Regulates *LBD16* Transcription and Promotes Adventitious Root Formation in *Arabidopsis* Leaf Explants"

_ijms, 2024, doi:10.3390/ijms252413340_

Round 1
Reviewer 1 Report
Comments and Suggestions for Authors
The manuscript "Depletion of Gibberellin Signaling Accelerates De novo Root Regeneration in Arabidopsis thaliana, which can Rescue Declined Adventitious Root Formation from erecta Mutant Leaf Explants" by Jing et al. (ijms-3287384) is a very interesting work concerning the analysis of GA signaling in the development of adventitious roots in Arabidopsis. The authors developed several experiments in different genetic backgrounds to analyze how gibberellins and the GA-synthesis blocking chemical Paclobutrazol can influence the formation of these organs. Transcriptomics analysis and GUS-staining were also used to identify candidate genes and check the role of LBD16, respectively, in the effects of PBZ. Despite the significant interest of this wok, I have several concerns that prevent me from recommending the publication of the manuscript in its present form.
First of all, I could not access the Supplementary files, and therefore could not check whether results and comments by the authors are properly supported. I do not know who is to blame for this issue, but I cannot accept the manuscript without that input.
Relevantly, the Results section is very confusing and reiterative. The authors seem to discuss the results here, what does not make sense. A complete re-working of this section is mandatory, providing only those references that are strictly needed, and leaving discussions and descriptions for the Discussion section.
Concerning the title, I seriously doubt that "depletion" is an accurate term for the effect of PBZ effect on GA signaling. PBZ blocks GA synthesis, but GA signaling keeps going on, indeed it may be stated that even more pronounced, although through the activity of DELLAs and feedback mechanisms. Relevantly, authors seem not to take into account that PBZ not only blocks GA synthesis, but also degradation of Abscisic Acid (Desta, B., Amare, G. Paclobutrazol as a plant growth regulator. Chem. Biol. Technol. Agric. 8, 1 (2021). https://doi.org/10.1186/s40538-020-00199-z), what might also be having significant effects on the rooting responses analyzed.
Other significant issue that I came across is the use of self-citations that might not be necessary. For instance, if needed, there are probably better references for grafting than the one used (7), or when describing the GUS-staining (123), which just redirects to 124. I understand these might be not deliberated mistakes, but should be corrected.
On the other hand, it is not clear if the induction of roots from detached leaves implies a wound. In that case, transcriptomics analysis should be interpreted taking that into account.
I will provide other comments in the order I found them on the manuscript:
- Keywords: I do not think abbreviations are needed in this section. Besides, LRP1 does not seem to be a relevant one, especially since it is not even mentioned in the results section.
- Line 35. What is grass family?
- Lines 56-60: auxin synthesis. There are other relevant routes for IAA synthesis from tryptophan, see Morffy and Strader 2020 (https://doi.org/10.1016/j.pbi.2020.02.002). Indeed, some of them and later mentioned by the authors (Lines 327-331).
- Line 86: there are probably more recent cites than this one. Similar issue with the cites in Line 96, please provide up-to-date cites.
- Lines 144-145: reiterative, and as already said, this is not Discussion.
- Line 150: Sorin et al., 2005. Why is this cite here?
- Line 168. Figure 1 C-E,D. Please correct.
- Line 174: 3 days in a row for a rooting experiment is not a time-course, moreover when almost all explants have already rooted.
- Lines 210-211: There are four families of auxin transporters, and NPA is only active on PINs.
- Lines 243-246: mutants display DNRR response, but not equally. Therefore something else is missing in the interpretation of the authors.
- Line 361: I do not think it was an additional treatment, I understand both compounds were placed together in the growing media. Concerning this Figure 5, expression can be detected at both mock treatment and GA+PBZ, what drives me to the previous question on whether the leaves were wounded at the beginning of the experiment. Besides, the signal seems to be detected away from the rooting point.
- Lines 400-401: How can an heterodimer have contrasting effects on the target genes?
- Lines 403-406: very confusing statement.
- Lines 418-425: first PBZ down-regulates DELLAs, then induces them. Please clarify or correct.
- Lines 466-483. the whole paragraph is speculative and confusing.
- Lines 578-579, and others: "more than...". Please state clearly the number of samples used.
- Lines 583-585: Did the authors used a different concentration of PBZ for the physiological and transcriptomics experiments?
- Bioinformatics: Please cite properly the tools used (FastQC, Hisat, DESeq2).
- Line 618: a more detailed description of these experiments is required.
- Figure 1, D-E-H: Scale bar is too small. Legend should be corrected for clarification purposes.
- Figure 4D: Are these the normalized counts from the RNA-seq? Please explain.
Overall, the authors have achieved relevant results that are interesting, but a re-writing of the manuscript is mandatory. I encourage them to consider these recommendations to get their valuables results published.
Comments on the Quality of English LanguageEnglish should be corrected, many times verbs are in present tense and should be in past tense.
Author Response
Reviewer 1:
Comments 1: The manuscript "Depletion of Gibberellin Signaling Accelerates De novo Root Regeneration in Arabidopsis thaliana, which can Rescue Declined Adventitious Root Formation from erecta Mutant Leaf Explants" by Jing et al. (ijms-3287384) is a very interesting work concerning the analysis of GA signaling in the development of adventitious roots in Arabidopsis. The authors developed several experiments in different genetic backgrounds to analyze how gibberellins and the GA-synthesis blocking chemical Paclobutrazol can influence the formation of these organs. Transcriptomics analysis and GUS-staining were also used to identify candidate genes and check the role of LBD16, respectively, in the effects of PBZ. Despite the significant interest of this wok, I have several concerns that prevent me from recommending the publication of the manuscript in its present form.
First of all, I could not access the Supplementary files, and therefore could not check whether results and comments by the authors are properly supported. I do not know who is to blame for this issue, but I cannot accept the manuscript without that input.
Response 1: We are very sorry for the inconvenience with the inaccessible Supplementary file (but it seems that the other reviewers did not have this problem), and we hope that IJMS was finally providing the Supplementary file to you so that you can fully access our research.
------
Comments 2: Relevantly, the Results section is very confusing and reiterative. The authors seem to discuss the results here, what does not make sense. A complete re-working of this section is mandatory, providing only those references that are strictly needed, and leaving discussions and descriptions for the Discussion section.
Response 2: Although we fully agree that introduction, results, and discussion should be clearly separated, they also should be readable without consulting the other sections of the paper. Therefore, the sections in the result part require short introductions with references and conclusions that can involve short discussions. This is particularly true if the narrative of a paper, such as our manuscript, is driven by controversies in the literature and require enormous background knowledge (with references), and on the other hand conclusions so that the reader can understand the decision for the design of the next/follow-up experiment in the results.
However, we carefully revised the result section to make it more accessible for the readers, while our changes also addressed the concerns raised by reviewer 1 above, then ever suitable. Please see the manuscript with track changes for details.
------
Comments 3: Concerning the title, I seriously doubt that "depletion" is an accurate term for the effect of PBZ effect on GA signaling. PBZ blocks GA synthesis, but GA signaling keeps going on, indeed it may be stated that even more pronounced, although through the activity of DELLAs and feedback mechanisms.
Response 3: Since two reviewer (reviewer 1 and reviewer 3) criticized the original title, we decided to change the title to “Depletion of Gibberellin Signaling Upregulates LBD16 Transcription and Promotes Adventitious Root Formation in Arabidopsis Leaf Explants”, although we are still thinking that the original title was sound. However, we kept the term “depletion”, since other terms, such as ‘loss’ (that is to closely associated with loss-of-function mutants) or ‘lack’ (that implies a total loss), do not fit any better. The meaning of “depletion” is just ‘reduction in the number or quantity of something’. And GA signaling experts and we are certain that the accumulation of DELLA protein in the dark through PBZ is a result of blocked GA synthesis which ultimately causes strongly reduced GA signaling (and GA response). We are not sure on what your opinion based that “PBZ blocks GA synthesis, but GA signaling keeps going on, indeed it may be stated that even more pronounced, although through the activity of DELLAs and feedback mechanisms.” Yes, DELLA are the targets of GA signaling, but high GA signaling results in DELLA degradation, blocking of GA signaling (which can be indirectly/upstream by blocking GA biosynthesis by PBZ) results in DELLA accumulation that blocks GA responses. Part of the GA responses is repressing of some of the GA (& BR) biosynthesis enzyme genes (this forms a negative feedback loop that prevent overactivation of GA signaling). With other words, PBZ treatment blocks the enzyme activity of GA3, which blocks GA biosynthesis in general, since GA3 catalyzes an essential but limiting step; consequently, GA content drops, which results in reduced GA signaling that results in accumulation of DELLAs that blocks GA responses including the repression of GA biosynthesis gene expression; consequently, some GA biosynthesis genes and their encoded enzymes are highly expressed that DOES NOT results in higher GA content, since the inhibited GA3 still CANNOT catalyze an essential but limiting step of GA biosynthesis.
------
Comments 4: Relevantly, authors seem not to take into account that PBZ not only blocks GA synthesis, but also degradation of Abscisic Acid (Desta, B., Amare, G. Paclobutrazol as a plant growth regulator. Chem. Biol. Technol. Agric. 8, 1 (2021). https://doi.org/10.1186/s40538-020-00199-z), what might also be having significant effects on the rooting responses analyzed.
Response 4: Although we fully agree and actually addressed the question that it is an interesting and important question, whether increased ABA content plays a role in the positive effects of PBZ on DNRR, we decided not to mention and/or discuss ABA in the first manuscript version in order not to confuse the reader. However, we checked first, whether the RNA seq data can give evidence for more or less ABA content. At first glance, the ABA content seemed increased in PBZ-treated er-105 leaf explants indicated by a significant higher overlap of ABA-regulated and PBZ-regulated genes, but most of these expression changes could be a result of less GA content as well (see new Supplementary Figure S4A). With other words, we think that although there is a significant overlap between PBZ- and ABA-regulated genes it is possibly an indirect result of depleted GA signaling and response (Supplementary Figure S4A). Furthermore, we now show that, even if PBZ would increases ABA content, ABA treatment inhibits AR formation (Supplementary Figure S4B), which is not in line with our observation that PBZ promotes AR formation. We entered a new paragraph in chapter 2.5 describing the problem, the control experiments, results, and our conclusions.
------
Comments 5: Other significant issue that I came across is the use of self-citations that might not be necessary. For instance, if needed, there are probably better references for grafting than the one used (7), or when describing the GUS-staining (123), which just redirects to 124. I understand these might be not deliberated mistakes, but should be corrected.
Response 5: We add another reverence for grafting, the Current Biology Primer article "Grafting" by Charles W. Melnyk and Elliot M. Meyerowitz, 2014. We removed the reference 123.
-------
Comments 6: On the other hand, it is not clear if the induction of roots from detached leaves implies a wound. In that case, transcriptomics analysis should be interpreted taking that into account.
Response 6: 1.) How exactly should any scientist be able to work on leaf explants, whose pure existence require cutting, but without wounding? 2.) It is well establish that DNRR require wounding see Zhang et al., 2019 and Ye et al., 2020. 3.) We clearly mention this fact with references in line 63-66: "To trigger earlier steps of auxin biosynthesis in detached Arabidopsis leaves, wounding-induced jasmonate signaling activates the ETHYLENE RESPONSE FACTORs ERF109 and ERF111, which in turn upregulates the auxin biosynthesis gene ANTHRANILATE SYNTHASE α1 (ASA1) [24], [25]." 4.) We mention "wound", "wounded" and "wounding" 26x in the manuscript. This includes terms such as "wounding-induced DNRR from leaf explants" in line 147 and 165-166. 5.) The comparison of our PBZ-regulated genes data "with wounding-regulated genes in leaf explants during DNRR" is part of our expression analysis, see line 310-319.
-------
Comments 7: I will provide other comments in the order I found them on the manuscript:
- Keywords: I do not think abbreviations are needed in this section. Besides, LRP1 does not seem to be a relevant one, especially since it is not even mentioned in the results section.
Response 7: We removed the abbreviations ARs, PAC, and GA. We kept DNRR and PBZ since these abbreviations are partally better known than their full name. We add the full name of LATERAL ORGAN BOUNDARIES DOMAIN 16 and LATERAL ROOT PRIMORDIUM 1. We add a sentence about LRP1 at the end of the result section: "Nevertheless, we found also other candidate genes such as LRP1, which could encode GA/BR-related DNRR factors working beyond and in parallel to LBD16 (Figure 4C-E)."
-------
Comments 8: Line 35. What is grass family?
Response 8: Just for the sake that any reader does not know that "grass family" is a synonym for 'Poaceae' we add this term in brakes behind "grass family".
-------
Comments 9: Lines 56-60: auxin synthesis. There are other relevant routes for IAA synthesis from tryptophan, see Morffy and Strader 2020 (https://doi.org/10.1016/j.pbi.2020.02.002). Indeed, some of them and later mentioned by the authors (Lines 327-331).
Response 9: We agree that there are other relevant routes for IAA synthesis from tryptophan, but they are not known to play a role in DNRR. And we mentioned the other auxin synthesis pathways then ever we found the gene expression of at least one of their components significantly changed. We are not sure what the point here is. It should be clear that we should not describe auxin pathways that neither play a role in DNRR nor was regulated by PBZ.
-------
Comments 10: Line 86: there are probably more recent cites than this one. Similar issue with the cites in Line 96, please provide up-to-date cites.
Response 10: In my whole scientific career (over 20 years), I never had that a reviewer asked for 'up-to-date' citations ('cite' is a verb!). The problem is normally that authors forget to mention the original references. However, Lange 1998 is a very fine review with more than 170 citations. And why are Busov et al. 2006, Mauriat et al. 2014, and Zhang et al. 2021 not 'up-to-date' citations?
-------
Comments 11: Lines 144-145: reiterative, and as already said, this is not Discussion.
Response 11: Yes, we fully agree that this is not discussion, but a short introduction (with references) to the issue in hand. In our opinion, this allows the readers to have an easier access to the research subject. In anyway, this is a question of taste and style and not of any rules.
-------
Comments 12: Line 150: Sorin et al., 2005. Why is this cite here?
Response 12: Because this reference described/used for the first time the "AR formation assays" that we used in our study.
-------
Comments 13: Line 168. Figure 1 C-E,D. Please correct.
Response 13: We correct it (exchanged D by H).
-------
Comments 14: Line 174: 3 days in a row for a rooting experiment is not a time-course, moreover when almost all explants have already rooted.
Response 14: If 3 days in a row are a time-course or not is, if at all, a matter of semantics. Yes, it is unfortunate that we did not measure the rooting rate at earlier time points. However, all relevant conclusions based on the end and not on the beginning of these time-courses.
-------
Comments 15: Lines 210-211: There are four families of auxin transporters, and NPA is only active on PINs.
Response 15: Yes, exactly. Then NPA (that blocks PINs) can blocked all DNRR activity but GA cannot, the effect of GA on auxin transport is less than the one of NPA that already do not block all auxin transport. Thus, our statement "GA might decrease auxin transport, but does not block the transport entirely" is fully correct
-------
Comments 16: Lines 243-246: mutants display DNRR response, but not equally. Therefore something else is missing in the interpretation of the authors.
Response 16: Yes, the mutants display significant DNRR response although the carry additionally a loss of ER function mutation that almost eliminate all DNRR response (the declined rooting activity makes er mutants to 'difficult-to-root' plants. For better understanding, we add the sentence "With other words, loss of GA synthesis and signaling (in ga1-4, ga5-1, and gai-1) can rescue declined AR formation in difficult-to-root leaf explants such as er mutants." All three GA mutants are strong, but do not display a total loss of GA signaling and response. They are defect at different steps of the GA pathway and their wild-type versions contribute with different proportions to this step, (e.g. GA20ox1/GA5 have two functional homologues in Arabidopsis GA20ox2/-3), and their defects are of different types (e.g. gai-1 expresses a dominant-negative version that blocks (partially) the function of the other four DELLA proteins). Therefore, if two of the mutants would rescue the er mutation equally that would be nothing more than coincidence.
-------
Comments 17a: Line 361: I do not think it was an additional treatment, I understand both compounds were placed together in the growing media.
Response 17: Ok, here is another matter of semantics. If you add GA additionally to PBZ (= GA & PBZ double treatment), the term 'additional GA treatment' is correct.
Comments 17b: Concerning this Figure 5, expression can be detected at both mock treatment and GA+PBZ, what drives me to the previous question on whether the leaves were wounded at the beginning of the experiment.
Response 17b: Yes, the leaves were wounded (they are explants that involve cutting! See also Response 6). And yes, expression can be detected at both mock treatment and GA+PBZ, but PBZ single treatment displayed much stronger expression.
Comments 17c: Besides, the signal seems to be detected away from the rooting point.
Response 17c: That is not simply not true. The rooting site in leaf explants is above the cutting site, exactly there we find LBD16 expression.
-------
Comments 18: Lines 400-401: How can an heterodimer have contrasting effects on the target genes?
Response 18: Where exactly do we say that the BZR1-PIF4 heterodimer have contrasting effects on the target genes? We say that each transcription factor, BZR1 and PIF4, have partially opposite effects on the expression of common target genes. Transcription factors, like all proteins can bind with different interacting partners or working even alone.
-------
Comments 19: Lines 403-406: very confusing statement.
Response 19: No, it is not.
-------
Comments 20: Lines 418-425: first PBZ down-regulates DELLAs, then induces them. Please clarify or correct.
Response 20: In the whole manuscript, there is not a single sentence that states that PBZ down-regulates DELLAs because that is not the case.
-------
Comments 21: Lines 466-483. the whole paragraph is speculative and confusing.
Response 21: Yes, the whole paragraph is speculative, since we are in the discussion section of the manuscript.
-------
Comments 22: Lines 578-579, and others: "more than...". Please state clearly the number of samples used.
Response 22: Since we used partially more than 6 plates (e.g. 8 plates) but never less our statement "more than six plates of each treatment" is perfectly correct. The same is true for "more than 10 explants per genotype".
-------
Comments 23: Lines 583-585: Did the authors used a different concentration of PBZ for the physiological and transcriptomics experiments?
Response 23: Sorry for the typo. For DNRR assays, the concentration of PBZ were always 5 µM.
-------
Comments 24: Bioinformatics: Please cite properly the tools used (FastQC, Hisat, DESeq2).
Response 24: Wie add the references below to the methods of the manuscript:
For FastQC: Wingett, S. W., & Andrews, S. (2018). FastQ Screen: A tool for multi-genome mapping and quality control of sequencing reads. F1000Res. 2018 Aug 24:7:1338.
doi: 10.12688/f1000research.15931.2. eCollection 2018.
For Hisat2: Kim D, Paggi JM, Park C, Bennett C, Salzberg SL. Graph-based genome alignment and genotyping with HISAT2 and HISAT-genotype. Nat Biotechnol. 2019 Aug;37(8):907-915. doi: 10.1038/s41587-019-0201-4. Epub 2019 Aug 2. PMID: 31375807; PMCID: PMC7605509.
For DESeq2: Love MI, Huber W, Anders S. Moderated estimation of fold change and dispersion for RNA-seq data with DESeq2. Genome Biol. 2014;15(12):550. doi: 10.1186/s13059-014-0550-8. PMID: 25516281; PMCID: PMC4302049.
-------
Comments 25: Line 618: a more detailed description of these experiments is required.
Response 25: We extended the last sentence that reads now "Venny 2.1 (https://bioinfogp.cnb.csic.es/tools/venny/index.html) was used for comparisons of gene lists: PBZ-regulated genes identified in this study were compared with published lists of DEG in response to hormones, GA (Goda et al. 2008; Ogawa et al. 2003; Ribeiro et al. 2012), BR (Goda et al. 2008; Sun et al. 2010), ABA (Goda et al. 2008), and auxin (Goda et al. 2008; Vain et al. 2019); and wounding-regulated genes during DNRR (Ye et al. 2020).
-------
Comments 26: Figure 1, D-E-H: Scale bar is too small. Legend should be corrected for clarification purposes.
Response 26: We changed the scale bar.
-------
Comments 27: Figure 4D: Are these the normalized counts from the RNA-seq? Please explain.
Response 27: Yes, we used the CPM values in Supplementary Tables S1 and S2
-------
Comments 28: Overall, the authors have achieved relevant results that are interesting, but a re-writing of the manuscript is mandatory. I encourage them to consider these recommendations to get their valuables results published.
Response 28: We are very thankful for this intensive review. We addressed all concerns raised by Reviewer 1 and we changed the manuscript accordantly, whenever it improved the manuscript.
-------
Comments 29: Comments on the Quality of English Language
English should be corrected, many times verbs are in present tense and should be in past tense.
Response 29: For stylistic purpose, verbs are in present tense in sentence that contain general statements, conclusions, or results of other studies (politeness demand that we take results of other studies as facts so long we do not have reasonable doubt). The description of our experiments and results should be always in past tense. We proofread the whole manuscript and changed the tense of the verb whenever it was not following the rules above.
-------
Reviewer 2 Report
Comments and Suggestions for Authors
This work presents an interesting demonstration of the positive functions of gibberellin acid biosynthesis inhibitor paclobutrazol PBZ on adventitious root (AR) and de novo root regeneration (DNRR) formation in Arabidopsis. In the introduction, the authors present the research background clearly, and the treatment with the GA inhibitor PBZ to study the formation of AR and DNRR is an innovation point, and to further clarify the mechanism of AR formation. This work can be improved by including and discussing some of the following topics:
1. The author has clearly introduced the research background, and this paragraph should be deleted or incorporated into the introduction. “In Arabidopsis, PBZ treatment was frequently employed as inhibitor of GA biosynthesis to investigate the role of GA signaling in plant development [53], [54], [55], [56], [57], [58], [59]. However, PBZ treatment was neither used during etiolation-induced AR formation from hypocotyl nor wounding-induced DNRR from leaf explants. In contrast, GA is known to inhibit adventitious rooting from hypocotyl in Arabidopsis [42].”
2. This paragraph should appear in the Materials and Methods. “Wild-type (Col-0) plants were germinated by incubating the seeds in light for 3 - 5 hours before being grown in dark for 3 days to induce hypocotyl elongation. Well- elongated and etiolated seedlings were transferred to various treatments under long day (LD) light conditions for 7 days, and then, the number of ARs were determined”
3. Line 192, although professionals are familiar with why plants are cultivating themselves in the dark, the principle should also be introduced.
4. Please briefly describe the operation method for measuring plant length with Image J software in the Materials and Methods.
5. In all figures, please correct the text to the Times New Roman.
6. Please introduce the meaning of “root capacity”.
7. Please add the detection instrument for Figure 1. C-D.
8. In the abstract, the PAC is mentioned as the gibberellin acid biosynthesis inhibitor paclobutrazol, but the PBZ is applied in the text, please explain the problem.
Author Response
Reviewer 2:
Comments 1: This work presents an interesting demonstration of the positive functions of gibberellin acid biosynthesis inhibitor paclobutrazol PBZ on adventitious root (AR) and de novo root regeneration (DNRR) formation in Arabidopsis. In the introduction, the authors present the research background clearly, and the treatment with the GA inhibitor PBZ to study the formation of AR and DNRR is an innovation point, and to further clarify the mechanism of AR formation. This work can be improved by including and discussing some of the following topics:
- The author has clearly introduced the research background, and this paragraph should be deleted or incorporated into the introduction. “In Arabidopsis, PBZ treatment was frequently employed as inhibitor of GA biosynthesis to investigate the role of GA signaling in plant development [53], [54], [55], [56], [57], [58], [59]. However, PBZ treatment was neither used during etiolation-induced AR formation from hypocotyl nor wounding-induced DNRR from leaf explants. In contrast, GA is known to inhibit adventitious rooting from hypocotyl in Arabidopsis [42].”
Response 1: We removed the first part of/re-wrote the first sentence since it is redundant with the introduction.
-------
Comments 2: This paragraph should appear in the Materials and Methods. “Wild-type (Col-0) plants were germinated by incubating the seeds in light for 3 - 5 hours before being grown in dark for 3 days to induce hypocotyl elongation. Well- elongated and etiolated seedlings were transferred to various treatments under long day (LD) light conditions for 7 days, and then, the number of ARs were determined”
Response 2: Yes, we agree and moved the paragraph into the Materials and Methods, 4.2. De novo root regeneration (DNRR) and hormone treatment.
-------
Comments 3: Line 192, although professionals are familiar with why plants are cultivating themselves in the dark, the principle should also be introduced.
Response 3: Here, we describe the problems with culturing leaf explant on GA plants in the dark. Why we used dark conditions is described in line 177 -179: During DNRR in the dark, auxin biosynthesis by YUC proteins is increased in comparison to light conditions accelerating the rooting rate in wild-type, ..."
-------
Comments 4: Please briefly describe the operation method for measuring plant length with Image J software in the Materials and Methods.
Response 4: We add "A reference with known length were employed to calibrate using the 'Set Scale' function. After calibration, all measurement results are automatically converted. After using the 'Freehand Line' tool to draw a line along the main root, the 'Measure' function were used and the measurement results (here, root length values) were displayed in the 'Results' window'."
-------
Comments 5: In all figures, please correct the text to the Times New Roman.
Response 5: 1.) The font in IJMS is 'Palatino linotype' and not 'Times New Roman'. 2.) Many journals such as JXB have different fonts in the figures in comparison to the main text. 3.) Even if we do not know the exact rule for figures in IJMS, the fonts in the figures in our last IJMS paper neither were 'Palatino linotype' nor 'Times New Roman' and that was fine with the typesetting editor of IJMS and us.
-------
Comments 6: Please introduce the meaning of “root capacity”.
Response 6: Sorry for the misspelling, it is "rooting capacity" and NOT “root capacity” or "regenerative capacity." We checked the whole manuscript and corrected these typos.
-------
Comments 7: Please add the detection instrument for Figure 1. C-D.
Response 7: We add "The photos were taken by a Nikon (SMZ25) microscope."
-------
Comments 8: In the abstract, the PAC is mentioned as the gibberellin acid biosynthesis inhibitor paclobutrazol, but the PBZ is applied in the text, please explain the problem.
Response 8: In some papers, the acronym of paclobutrazol is 'PAC'; however, 'PBZ' is the more common abbreviation and so we removed now 'PAC' from the whole manuscript.
-------
Reviewer 3:
Comments 1: The article with the title “Depletion of Gibberellin Signaling Accelerates De novo Root Regeneration in Arabidopsis thaliana, which can Rescue Declined Adventitious Root Formation from erecta Mutant Leaf Explants”is very heavy and incomprehesive.
First the title is confusing, please change the title in such a way to be consistent with the manuscript text.
Response 1: Since two reviewer (reviewer 1 and reviewer 3) criticized the original title, we decided to change the title to “Depletion of Gibberellin Signaling Upregulates LBD16 Transcription and Promotes Adventitious Root Formation in Arabidopsis Leaf Explants”, although we are still thinking that the original title was sound as well.
-------
Comments 2: The abstract looks like a summary for a review type manuscript. The aim was not declared also I am not sure if some original results are there ort not.
Response 2: We agree and accordantly revised the abstract to make it more accessible for the readers. The aim is now declared: "In order to clarify this contradiction, we employed the GA biosynthesis inhibitor paclobutrazol (PBZ) and found that …" Due to the limited space, we moved the last sentence to the end of the introduction.
-------
Comments 3: Too many abbreviations in the abstract and also in the keywords only the firsy time a abbreviation is used you have to write the full name. Please see AR in the abstract and keywords.
Response 3: We fully agree that only if an abbreviation is used the first time, you have to write the full name; and you used afterwards only the abbreviation. However, if you do so in the abstract or title, you have still to write the full name of the abbreviation then it is used the first time in the main text. We revised the manuscript accordantly. On the other hand, we think that readers can handle the 4 abbreviations (AR, DNRR, GA, LBD16) in the abstract, especially since these four one are also the main abbreviations in the main text. Nevertheless, we removed ARs, GA, PAC from the key words.
-------
Comments 4: Do not uppercase words or other gathering words. Please see rows 38-39, 46, 57 and so on in the entire text especially from the introduction section.
Response 4: Gene and protein names, such as LATERAL ORGAN BOUNDARIES DOMAIN and AUXIN RESPONSE FACTOR, in Arabidopsis have accordantly to the Community Standards for Arabidopsis Genetics (Meinke and Koornneef, 1997, The Plant Journal) to be uppercased (and gene names additionaly italic).
-------
Comments 5: Please consult carefully the instructions for authors template, the citation not seem corectly done row 96 for example (114, 118)
Response 5: We did not find the incorrect citations mentioned above in our manuscript.
-------
Comments 6: Why are not the figures inserted on their place?
Response 6: Because IJMS accepts free format submission. We now moved the figures to the right places accordantly to the instruction for the authors.
-------
Comments 7: Please separate the results and discuttion section and treat them standing alone. Remove the citations from the results.
Response 7: We agree that some sentences carry too many citations. So, we removed the first sentence in line 144-146 entirely. However, references in the results are good scientific practice. The corresponding authors used citations in the results of e.g. three of his Plant Cell papers. We think that is good enough for The Plant Cell should be good enough for IJMS. Although we fully agree that results and discussion should be clearly separated, they also should be readable without consulting the other sections of the paper. Therefore, the sections in the result part require short introductions with references and conclusions that can involve short discussions. This is particularly true if the narrative of a paper, such as our manuscript, is driven by controversies in the literature and require enormous background knowledge with references, and on the other hand conclusions so that the reader can understand the decision for the design of the next/follow-up experiment in the results.
-------
Reviewer 3 Report
Comments and Suggestions for Authors
The article with the title “Depletion of Gibberellin Signaling Accelerates De novo Root Regeneration in Arabidopsis thaliana, which can Rescue Declined Adventitious Root Formation from erecta Mutant Leaf Explants”is very heavy and incomprehesive.
First the title is confusing, please change the title in such a way to be consistent with the manuscript text.
The abstract looks like a summary for a review type manuscript. The aim was not declared also I am not sure if some original results are there ort not.
Too many abbreviations in the abstract and also in the keywords only the firsy time a abbreviation is used you have to write the full name. Please see AR in the abstract and keywords.
Do not uppercase words or other gathering words. Please see rows 38-39, 46, 57 and so on in the entire text especially from the introduction section.
Please consult carefully the instructions for authors template, the citation not seem corectly done row 96 for example (114, 118)
Why are not the figures inserted on their place?
Please separate the results and discuttion section and treat them standing alone. Remove the citations from the results.
Author Response
Reviewer 3:
Comments 1: The article with the title “Depletion of Gibberellin Signaling Accelerates De novo Root Regeneration in Arabidopsis thaliana, which can Rescue Declined Adventitious Root Formation from erecta Mutant Leaf Explants”is very heavy and incomprehesive.
First the title is confusing, please change the title in such a way to be consistent with the manuscript text.
Response 1: Since two reviewer (reviewer 1 and reviewer 3) criticized the original title, we decided to change the title to “Depletion of Gibberellin Signaling Upregulates LBD16 Transcription and Promotes Adventitious Root Formation in Arabidopsis Leaf Explants”, although we are still thinking that the original title was sound as well.
-------
Comments 2: The abstract looks like a summary for a review type manuscript. The aim was not declared also I am not sure if some original results are there ort not.
Response 2: We agree and accordantly revised the abstract to make it more accessible for the readers. The aim is now declared: "In order to clarify this contradiction, we employed the GA biosynthesis inhibitor paclobutrazol (PBZ) and found that …" Due to the limited space, we moved the last sentence to the end of the introduction.
-------
Comments 3: Too many abbreviations in the abstract and also in the keywords only the firsy time a abbreviation is used you have to write the full name. Please see AR in the abstract and keywords.
Response 3: We fully agree that only if an abbreviation is used the first time, you have to write the full name; and you used afterwards only the abbreviation. However, if you do so in the abstract or title, you have still to write the full name of the abbreviation then it is used the first time in the main text. We revised the manuscript accordantly. On the other hand, we think that readers can handle the 4 abbreviations (AR, DNRR, GA, LBD16) in the abstract, especially since these four one are also the main abbreviations in the main text. Nevertheless, we removed ARs, GA, PAC from the key words.
-------
Comments 4: Do not uppercase words or other gathering words. Please see rows 38-39, 46, 57 and so on in the entire text especially from the introduction section.
Response 4: Gene and protein names, such as LATERAL ORGAN BOUNDARIES DOMAIN and AUXIN RESPONSE FACTOR, in Arabidopsis have accordantly to the Community Standards for Arabidopsis Genetics (Meinke and Koornneef, 1997, The Plant Journal) to be uppercased (and gene names additionaly italic).
-------
Comments 5: Please consult carefully the instructions for authors template, the citation not seem corectly done row 96 for example (114, 118)
Response 5: We did not find the incorrect citations mentioned above in our manuscript.
-------
Comments 6: Why are not the figures inserted on their place?
Response 6: Because IJMS accepts free format submission. We now moved the figures to the right places accordantly to the instruction for the authors.
-------
Comments 7: Please separate the results and discuttion section and treat them standing alone. Remove the citations from the results.
Response 7: We agree that some sentences carry too many citations. So, we removed the first sentence in line 144-146 entirely. However, references in the results are good scientific practice. The corresponding authors used citations in the results of e.g. three of his Plant Cell papers. We think that is good enough for The Plant Cell should be good enough for IJMS. Although we fully agree that results and discussion should be clearly separated, they also should be readable without consulting the other sections of the paper. Therefore, the sections in the result part require short introductions with references and conclusions that can involve short discussions. This is particularly true if the narrative of a paper, such as our manuscript, is driven by controversies in the literature and require enormous background knowledge with references, and on the other hand conclusions so that the reader can understand the decision for the design of the next/follow-up experiment in the results.
-------
Round 2
Reviewer 2 Report
Comments and Suggestions for Authors I recommend accepting.